# Secondary-Type Mutations in Acute Myeloid Leukemia: Updates from ELN 2022

**DOI:** 10.3390/cancers15133292

**Published:** 2023-06-22

**Authors:** Ian M. Bouligny, Keri R. Maher, Steven Grant

**Affiliations:** Division of Hematology and Oncology, Department of Internal Medicine, Virginia Commonwealth University Massey Cancer Center, Richmond, VA 23298, USA; keri.maher@vcuhealth.org (K.R.M.); steven.grant@vcuhealth.org (S.G.)

**Keywords:** *SF3B1*, *SRSF2*, *U2AF1*, *ZRSR2*, *BCOR*, *EZH2*, *STAG2*, *RUNX1*, *ASXL1*, hematological malignancies, acute myeloid leukemia

## Abstract

**Simple Summary:**

Therapeutic advances in acute myeloid leukemia (AML) are dependent on identifying and targeting the molecular aberrations that drive disease. The European LeukemiaNet (ELN) 2022 guidelines have improved the categorization of AML into distinct molecular subgroups. One of the most notable recent inclusions is a group of mutations highly specific for secondary AML. This review examines how each of the secondary-type mutations contributes to the genesis of leukemia while spotlighting potential therapeutic avenues. While we highlight the limitations of the ELN 2022 revision, we also emphasize current progress, recent breakthroughs, and novel therapeutic options tailored to each molecular subset. This review provides a background and foundation for rational molecular-based therapeutic approaches and combination strategies to inspire future clinical trial designs.

**Abstract:**

The characterization of the molecular landscape and the advent of targeted therapies have defined a new era in the prognostication and treatment of acute myeloid leukemia. Recent revisions in the European LeukemiaNet 2022 guidelines have refined the molecular, cytogenetic, and treatment-related boundaries between myelodysplastic neoplasms (MDS) and AML. This review details the molecular mechanisms and cellular pathways of myeloid maturation aberrancies contributing to dysplasia and leukemogenesis, focusing on recent molecular categories introduced in ELN 2022. We provide insights into novel and rational therapeutic combination strategies that exploit mechanisms of leukemogenesis, highlighting the underpinnings of splicing factors, the cohesin complex, and chromatin remodeling. Areas of interest for future research are summarized, and we emphasize approaches designed to advance existing treatment strategies.

## 1. Introduction

Next-generation sequencing has transformed the landscape of acute myeloid leukemia. Over the last several decades, the identification of recurrent genetic abnormalities in AML has guided modern clinical approaches toward a molecular era. The recent 2022 revision to the European LeukemiaNet guidelines builds upon its predecessor [1,2]; in addition to refinement of the favorable- and intermediate-risk categories, seven new mutations joined the adverse-risk category: *SF3B1*, *SRSF2*, *U2AF1*, *ZRSR2*, *BCOR*, *EZH2*, and *STAG2*. These seven mutations, along with several others, comprise a group known as the secondary-type mutations, aptly named for their ubiquity in secondary AML [3].

Secondary AML defines a subset of the disease with notoriously adverse outcomes. Based on preceding myelodysplastic neoplasms (MDS), myeloproliferative neoplasms (MPNs), or therapy-related clonal aberrations, secondary AML is associated with lower remission rates and overall survival compared with de novo AML [4,5,6]. While secondary-type mutations are frequently harbingers of dismal outcomes, the relationship between the presence of these mutations and prognosis is more complex. For example, the ELN 2022 revision classifies secondary-type mutations as adverse-risk AML but emphasizes that the risk should not be considered adverse if secondary-type signatures co-occur with favorable-risk mutations [2]. Indeed, real-world analyses from our center and others demonstrated that AML with secondary-type mutations and co-mutated *NPM1* or rearrangements of the core-binding factor retain their favorable prognoses [7,8]. In contrast, the effects of mutational cooperation between secondary-type mutations and other mutations in the intermediate- and adverse-risk categories are less clearly defined. If our goal is to adopt a tailored approach to therapy for this heterogenous cohort of AML, what can be done for those with secondary-type mutations, and how impactful would such approaches be?

Spliceosome mutations, including *SF3B1, SRSF2, U2AF1,* and *ZRSR2*, are found in approximately 65% of patients with MDS [9]; they are among the earliest mutations to occur [10]. The acquisition of additional somatic lesions confers a subclonal survival advantage, driving the progression from MDS to AML [11]. In the VCU Massey Cancer Center Project ERIS database involving 600 patients with AML, 28.9% harbored a spliceosome mutation or a mutation in *BCOR*, *EZH2*, or *STAG2;* 36.8% harbored mutations in any of the seven previously mentioned genes or *ASXL1* or *RUNX1* (Figure 1). Alongside AML with mutated *NPM1*, the chromatin-spliceosome subgroup is one of the largest subgroups of AML [12]. Consequently, the impact of novel targeted therapies or additional treatment strategies unique to AML with secondary-type mutations would be substantial and would represent significant progress for many patients. Thus, this review highlights the mechanisms of leukemogenesis and potential targets of each of the seven secondary-type mutations introduced in the ELN 2022 revision—in addition to *ASXL1* and *RUNX1*. First, we review the spliceosome to appreciate the contributions of each mutation to the genesis of AML.

### The Spliceosome and AML

The spliceosome coordinates mRNA splicing—the conversion of pre-mRNA into mature mRNA [13]. The spliceosome is a gargantuan complex composed of hundreds of proteins and small nuclear RNAs (snRNAs). When snRNAs bind to proteins, they form small nuclear ribonucleoproteins (snRNPs). These snRNPs comprise the major spliceosome—responsible for 99.5% of human splicing—and the minor spliceosome, which splices the remaining 0.5% [14]. Spliceosomes create mature mRNA from pre-mRNA by removing non-coding sequences called introns, named because they are intragenic regions. Introns are sandwiched between a proximal 5′ splice site and a distal 3′ splice site. To remove the introns, uridine-rich snRNPs introduce the 5′ splice site to the 3′ splice site, forming an intron loop called a lariat. The 5′ splice site attacks the 3′ splice site, excising the intron-containing lariat and leaving behind mature mRNA [15]. In summary, splicing produces a mature mRNA transcript—a template for protein translation (Figure 2).

To appreciate the intricacies of the most common spliceosome mutations in AML, we will focus on the major spliceosome’s key players, the five uridine (U)-rich snRNPs: U1, U2, U4, U5, and U6. Starting from the proximal end of a target splice region, U1 binds to the 5′ splice site. Next, the budding spliceosome locates the distal end of a splice region through a two-step process. First, splicing factor 1 (SF1) binds to the branch site, an area near the 3′ splice site [16]. Second, the U2 axillary factors (U2AFs) form a complex that tethers to the 3′ splice site adjacent to SF1. After recruiting SF1 and the U2AF complex, U2 snRNPs displace SF1 from the branch site [16]. Finally, a triplet complex consisting of the remaining snRNPs—U4, U5, and U6—joins U1, U2, and the U2AF complex to create the activated spliceosome [17]. The spliceosome excises the intron-containing lariat and ligates the exons through two sequential transesterification reactions.

The diversity of proteins created through splicing is staggering; it is directly related to the selection of splice sites. The spliceosome always recognizes some splice sites, known as constitutive splice sites. In contrast, the recognition of other splice sites depends on the recruitment of additional factors; therefore, they are only occasionally spliced. These are alternative splice sites, which create combinatorial effects on protein diversity [11,17]. In multi-exon genes, 5′ splice sites can join different 3′ splice sites, and 3′ splice sites can join different 5′ splice sites, resulting in a large array of context-specific combinations achieved by alternative splicing. The serine/arginine-rich (SR) proteins and the heterogeneous nuclear ribonucleoproteins (hnRNPs) control alternative splicing by respectively promoting or repressing splicing events [18]; therefore, the regulatory SR proteins and hnRNPs can be oncoproteins or tumor suppressors. Consequently, mutations in spliceosome proteins or their regulators are frequent events in AML.

The four most commonly mutated spliceosome mutations in AML are *SF3B1*, *SRSF2*, *U2AF1*, and *ZRSR2*. The first three are components of the major spliceosome; they exist in a heterozygous state and usually have altered or gain-of-function effects—consistent with their role as oncoproteins. In contrast, *ZRSR2* is a component of the minor spliceosome and commonly exhibits a loss-of-function effect, acting as a tumor suppressor [17]. The spliceosome mutations are largely mutually exclusive and exhibit synthetic lethality [19]. This fundamental principle suggests that leukemic cells with spliceosome mutations may be uniquely sensitive to additional disruptions in splicing [20]. Thus, we will explore how the most common spliceosome mutations contribute to leukemogenesis and the avenues that may lead to more personalized therapeutic approaches.

## 2. Aberrations of Spliceosome Genes

### 2.1. SF3B1

Splicing factor 3B subunit 1, also known as SF3B1, is a protein component of the U2 snRNP; it binds to the branch site adjacent to the 3′ splice site. Therefore, the function of SF3B1 is to assist in correctly localizing the spliceosome to the 3′ splice site. In Project ERIS, mutations in *SF3B1* occurred in 2.6% (95% CI, 1.4 to 4.9) of patients with AML, with a median upfront variant allele frequency (VAF) of 46.1%, consistent with its known heterozygous presentation; additional aggregate analyses have confirmed *SF3B1*^mut^ frequencies of 2–5% in AML [21]. Mutations in *SF3B1* frequently guide the spliceosome to an alternative 3′ splice site. Consequently, the introduction of a premature stop codon in target gene expression due to *SF3B1*^mut^ is a common occurrence, resulting in reduced protein expression due to aberrant splicing [22].

The impact of *SF3B1*^mut^ in hematological malignancies can be best appreciated in MDS with ringed sideroblasts. The abnormal splice site degrades a mitochondrial iron exporter, ABCB7 [23]. Reduction in functional ABCB7 leads to mitochondrial iron retention in erythroblasts, characteristically resulting in ringed sideroblasts, aberrant erythropoiesis, and a predilection for anemia [21]. Other mutations in *SF3B1* create terminal blocks in erythroid maturation but without ringed sideroblasts [9,24]. In AML, *SF3B1* mutations are frequently found with AML-driving genes: *RUNX1*^mut^ or rearrangements of *MECOM* [25]. As *SF3B1* mutations are often early events, this finding suggests that AML with *SF3B1*^mut^ arises following the acquisition of additional adverse-risk mutations—highlighting the need for more effective therapeutic approaches.

The numerous inhibitors of SF3b are the most well-studied spliceosome inhibitors. Pladienolide analogs were the first splicing modulators to enter clinical trials. One of the earliest splicing modulators was E7107, which disrupts splicing by targeting SF3b and inhibiting U2 [26,27]. Attributable to its activity in splicing interference, E7107 promotes apoptosis and cell cycle arrest [28,29]. While E7107 showed promising activity in pre-clinical studies, it was associated with vision loss and little efficacy in humans, resulting in trial termination [30]. This led to the investigation of other spliceosome inhibitors and the discovery of H3B-8800, a synthetic pladienolide derivative with a similar mechanism. A multi-center phase I trial evaluated H3B-8800 in MDS, CMML, and AML; while the drug was associated with decreased transfusion requirements and splicing modulation, it did not produce clinical responses [31,32].

Newer pre-clinical studies suggest that AML with internal tandem duplications in FMS-like tyrosine kinase (*FLT3*-ITD) may be more sensitive to the pladienolide analogs. In AML with *FLT3*-ITD, investigators noted increased induction of cell cycle arrest and a shift in the splicing patterns toward a pro-apoptotic state due to reduced *MCL1* following pladienolide treatment [33]. These findings suggest a biological rationale for combining splicing modulators with FLT3 inhibitors or inducers of apoptosis, such as venetoclax.

### 2.2. SRSF2

Serine/arginine-rich splicing factor 2 (SRSF2) is a regulatory SR protein that promotes splicing events by binding to splicing enhancer segments [34]. Mutated SRSF2 has reduced binding to splicing enhancers, culminating in global splicing changes [16]. Mutations in *SRSF2* result in the mis-splicing of *EZH2* and *BCOR*, explaining the observed mutual exclusivity between these genes [35,36]. Due to SRSF2^mut^-induced splicing, alterations in splicing factors block hematopoietic differentiation, which impairs maturation and drives leukemogenesis [37].

Mutations in *SRSF2* ubiquitously occur at proline 95, identified in 90.9% (95% CI, 76.4 to 96.9) of AML in Project ERIS—the reminder commonly appear at proline 94 or 96. A single allele mutation in *SRSF2* at proline 95 disrupts RNA-binding specificity and is sufficient to induce myelodysplasia [35]. Consequently, a single mutation at proline 95 produced the phenotypic characteristics of MDS in mice: anemia, leukopenia, and dysplasia. These observations led investigators to assess the efficacy of spliceosome inhibitors in *SRSF2*^mut^ AML.

Both investigational drugs that interfere with SF3b were evaluated in *SRSF2*^mut^ AML. Investigators administered E7107 to mice with *SRSF2*^P95H^. Untreated mice developed bone marrow failure; treated mice had a significantly reduced leukemic burden [20]. Moreover, splicing inhibition was greater in *SRSF2*^mut^ AML than in wild-type genotypes [20]. Similarly, H3B-8800—a synthetic pladienolide derivative—also showed decreased leukemic burden in *SRSF2*^P95H^ mice compared with wild-type mice [38]. Neither drug, however, produced clinically significant activity in trials, leading to the investigation of alternative approaches.

The discovery that spliceosome proteins require phosphorylation for nuclear transport—a process regulated by the CDC-like kinase (CLK) family—paved the way for innovation [39]. CTX-712, an orally available CLK inhibitor, strongly inhibits the phosphorylation of SR proteins that bind SRSF2. An *SRSF2*^mut^ PDX model showed a significant and dose-dependent response to CTX-712; additionally, the drug demonstrated anti-leukemic efficacy in PDX AML models without spliceosome mutations [40]. In a phase I trial, CTX-712 was associated with a composite complete remission rate of 60.0% in AML, with further safety assessment ongoing [41]. A phase I/II trial assessing CTX-712 in AML and MDS is currently recruiting at MD Anderson Cancer Center (NCT05732103).

### 2.3. U2AF1

U2 small nuclear RNA axillary factor 1 (U2AF1) recognizes the 3′ splice site and recruits the U2 snRNP during splicing [42]. Mutations in U2AF1 induce splicing alterations in genes critical for hematopoietic stem cell function [43], predisposing stem cells to DNA damage. Therefore, a crucial role of U2AF1 is to maintain the function and regenerative potential of stem cells in hematopoiesis. Indeed, mouse models with *U2AF1* knockout show pancytopenia, a reduction in hematopoietic stem cells, and decreased survival [43].

U2AF1 has additional roles aside from splicing. U2AF1 and its binding partner, U2AF2, bind to cytoplasmic RNA and act as translational repressors [44]. *U2AF1*^S34F^ predisposes to increased translation of interleukin 8, upregulating the inflammatory response and correlating with a higher incidence of relapsed or refractory AML in humans; blocking interleukin 8 led to tumor burden reduction in mice [44]. Consistent with these early observations of an upregulated inflammatory state, the focus has shifted to interleukin-1-associated kinase 4 (IRAK4), the predominant isoform that arises from alternative splicing in MDS and AML (Figure 3). *U2AF1*^mut^ is responsible for the alternative splicing of IRAK4, which results in the retention of an exon and creates a long isoform: IRAK4-long (IRAK4-L).

Investigators discovered that inhibition of IRAK4-L disrupts leukemic growth [45]. Thus, the presence of *U2AF1*^mut^ in AML induces the expression of a novel therapeutic target, IRAK4-L, which is inhibited with emavusertib. Single-agent emavusertib demonstrated a composite complete remission rate of 40.0% in AML with spliceosome mutations, and 57.1% of high-risk MDS patients had <5% marrow blasts [46]. Emavusertib continues to be evaluated in multicenter clinical trials as monotherapy and in combination with azacitidine and venetoclax (NCT04278768).

### 2.4. ZRSR2

The zinc finger (CCCH type), RNA-binding motif, and serine/arginine rich 2 (ZRSR2) gene encodes a key member of the minor spliceosome that recognizes 3′-intron splice sites [47]. *ZRSR2* mutations differ from the remaining spliceosome mutations in two critical ways. First, *ZRSR2* mutations primarily affect the minor spliceosome; *SF3B1*, *SRSF2*, and *U2AF1* mutations tend to affect the major spliceosome. Second, mutations in *ZRSR2* are scattered throughout the length of the gene, in contrast to the other three spliceosome mutations, which occur at hotspots. As *ZRSR2* is located on the X chromosome, mutations are linked to male-predominant leukemia—particularly in blastic plasmacytoid dendritic cell neoplasm [48]. In the Project ERIS database, 75% (95% CI, 40.9 to 95.6) of patients with *ZRSR2*^mut^ were male, suggesting a similar male-dominant presentation in AML.

ZRSR2 is required for splice site recognition of U12-type introns, which are removed by the minor spliceosome. Knockdown of *ZRSR2* results in reduced differentiation of hematopoietic precursors due to the aberrant retention of U12-type introns, suggesting that a competent minor spliceosome is necessary for myeloid differentiation [47]. Loss of functional *ZRSR2* leads to increased self-renewal of hematopoietic stem cells [49]. Additionally, leukemia cells with *ZRSR2* knockdown exhibited slower growth compared with wild-type controls, a pattern not exclusive to loss of *ZRSR2*; *U2AF1*^mut^ AML demonstrated similar findings of reduced hematopoietic reconstitution [47]. Therefore, mutations in *ZRSR2* appear to lead to constitutive stem cell self-renewal but do not appear to contribute to proliferative phenotypes.

There is relatively limited data on adopting a targeted approach to *ZRSR2*^mut^ AML. Novel agents that reverse the retention of U12-type introns and restore the function of the minor spliceosome have yet to be widely identified or studied, representing an area of unmet clinical need and a growing area of research interest. However, several mis-spliced targets may be of interest—*ZRSR2* mutations frequently result in the aberrant splicing of several E2F and MAP kinase signaling regulators, including the RAS pathway [47]. Breakthroughs in the treatment of *ZRSR2*^mut^ AML will likely require the identification of compounds that restore the function of the minor spliceosome, induce synthetic lethality through inhibition of other targets, or modify downstream mediators dysregulated by aberrant U12-type intron splicing.

## 3. Aberrations of Chromatin Modifiers

### 3.1. BCOR

The BCL6 corepressor (*BCOR*) is a transcription factor that regulates stem cell function and hematopoiesis. To understand the role of BCOR in AML, we will first review the polycomb group proteins. The polycomb group proteins are responsible for remodeling chromatin; they are recognized for silencing *HOX* genes through chromatin structure alteration, a critical process for hematopoiesis. Two key complexes in the polycomb group family are polycomb repressive complex 1 (PRC1) and polycomb repressive complex 2 (PRC2). PRC1 catalyzes the ubiquitination of histones (a histone ubiquitin ligase), repressing target genes that control stem cell pluripotency [50]. PRC2 is a histone methyltransferase that inhibits transcription through chromatin compaction [51]. In the context of *BCOR*^mut^ AML, we will shift our focus to the interaction between PRC1 and PCR2.

BCOR is a key component of PRC1.1, a noncanonical PRC1 [52]. BCOR recruits PRC1.1 to specific chromatin sites, resulting in histone ubiquitination and subsequent recruitment of PRC2, culminating in transcriptional repression [52]. Loss of BCOR, therefore, results in decreased histone ubiquitination at the *HOX* and *CEBPA* promoters, which aberrantly activates these myeloid-related genes [53]. BCOR mutations consequently lead to the expansion of myeloid precursors and produce phenotypes characteristic of MDS, driving ineffective hematopoiesis and dysplasia [53]. Furthermore, in *BCOR*^mut^ MDS, the acquisition of proliferative mutations through clonal evolution promotes the progression of MDS to AML, commonly through NRAS, KRAS, or FLT3-ITD [3].

*BCOR* mutations are associated with primary refractory AML [54], frequently cooperating with oncogenic *KRAS* mutations—particularly *KRAS*^G12D^ [55]. In Project ERIS, *BCOR*^mut^ AML also commonly cooperated with *NRAS* mutations, most frequently at glycine 12, with a median VAF of 25.03% (range, 19.59 to 37.22). In striking accordance with these observations, *BCOR*^mut^ AML appears to be sensitive to tyrosine kinase inhibition targeting pathways involved in stem cell pluripotency [56]. Ongoing studies now focus on modulating signal transduction to overcome *BCOR*^mut^-mediated disease refractoriness in AML [56].

### 3.2. EZH2

As we have reviewed the role of chromatin compaction with BCOR in PRC1, we will similarly unpack the function of enhancer of zeste homologue 2 (EZH2) in PRC2. EZH2 is the catalytic component of PRC2, a transcriptional corepressor with lysine methyltransferase activity that controls gene expression through histone modification [51]. The addition of a methyl group to lysines on histones induces a repressive chromatin state. Mutations in *EZH2* are a double-edged sword: as its target genes are required for stem cell maintenance, both gain- and loss-of-function mutations result in characteristic hematopoietic perturbations. A detailed review of epigenetics in leukemogenesis can be found here [51].

EZH2 acts as an oncoprotein in gain-of-function mutations. Well-studied in the context of lymphomagenesis, EZH2 overexpression results in transcriptional suppression of genes responsible for differentiation [57]. During disease maintenance in AML, EZH2 adopts an oncogenic role, suggesting its function can be therapeutically targeted [51,58]. In this context, gain-of-function mutations in EZH2 can be targeted by EZH2 inhibitors such as valemetostat, tazemetostat, GSK126, or 3-deazaneplanocin. Of recent interest, valemetostat, a dual inhibitor of EZH1 and EZH2, recruits leukemic stem cells into the cell cycle while potentiating apoptosis following treatment with a hypomethylating agent and venetoclax [59]. Another combination approach appears viable with intensive chemotherapy: inhibition of EZH2 rendered AML cells more susceptible to anthracycline-based therapy, with relatively lower anthracycline doses producing marked leukemia suppression [60].

In contrast, loss-of-function EZH2 mutations are more challenging to approach. As gain-of-function mutations act as oncoproteins, loss-of-function EZH2 mutations, conversely, act as tumor suppressors. In mouse models, loss-of-function EZH2 mutations activate bivalent promoters that accelerate AML induction [58]. Therefore, the approach to loss-of-function *EZH2*^mut^ will require more innovative approaches compared with its gain-of-function counterpart.

Similar to *BCOR*^mut^ AML, loss-of-function of EZH2 is associated with resistance to cytarabine [61]; however, the mechanisms of resistance are complex—driven by apoptosis evasion, increased proliferation, and alteration of transporter function. We speculate that therapies targeting the control of apoptosis and proliferation may provide insights into overcoming *EZH2*^mut^-mediated resistance to treatment. An additional therapeutic avenue involves potentially targeting a mutually exclusive mutation. For example, *SRSF2*^mut^ and *EZH2* loss-of-function mutations are mutually exclusive in MDS [9]. Thus, in the context of loss-of-function *EZH2* mutations, SRSF2 inhibition theoretically induces synthetic lethality. Further research is needed to characterize the dynamic landscape associated with EZH2 loss-of-function in AML.

### 3.3. ASXL1

Additional sex combs-like 1 (ASXL1) is jointly responsible for deubiquitinating histones—a function it performs with its histone deubiquitinase partner, BRCA1-associated protein 1 (BAP1). BAP1 is part of the polycomb repressive deubiquitinase (PR-DUB) complex, which reverses the repressive ubiquitination signatures of polycomb repressive complex 1. Therefore, the PR-DUB complex promotes gene expression [50].

ASXL1 activates BAP1 [62]. In turn, the PR-DUB complex is recruited to chromatin and regulates the gene expression of critical mediators of hematopoietic development, including the *HOX* genes. Mutations in ASXL1 result in BAP1 hyperactivation. BAP1 gain-of-function creates a state of histone deubiquitination at *HOX* gene regions, leading to leukemic transformation [63]. Under normal conditions, PRC1 maintains the inactive state of *HOX* genes through histone ubiquitination at these regions; however, hyperactive BAP1 reverses PRC1-mediated ubiquitination signatures and creates constitutive *HOX* gene expression, resulting in myeloid leukemogenesis [64]. Therefore, BAP1 is a tumor promoter in *ASXL1*^mut^ AML [63].

Additionally, mutated ASXL1 acquires a novel binding partner: bromodomain-containing protein 4 (BRD4) [65]. BRD4 acetylates histones and activates gene transcription via RNA polymerase II, resulting in stem cell self-renewal [65]. ASXL1 mutations that bind to BRD4 have higher expression of genes responsible for leukemogenesis, suggesting that inhibition of the ASXL1 and BRD4 interaction potentiates leukemic cell death [65].

Indeed, small-molecule inhibitors that target bromodomain and extraterminal (BET) domain proteins—including BRD4—demonstrate antileukemic activity [66]. BRD4 inhibition reduced the expression of self-renewal genes in *ASXL1*^mut^ AML by inhibiting the interaction between mutated ASXL1 and BRD4, suggesting BET inhibitors may have a role in *ASXL1*^mut^ AML [65]. Furthermore, BET inhibitors synergize with venetoclax in PDX models and may even help overcome venetoclax resistance, paving the way for combination approaches [67,68]. Other approaches include BAP1 inhibitors for *ASXL1*^mut^ AML, which were shown to inhibit *HOX* gene expression [63].

### 3.4. STAG2

The cohesin subunit SA-2 is encoded by *STAG2*, one of the four core units of the cohesin complex. The cohesin complex regulates chromatin structure; it is named because it mediates the cohesion of sister chromatids during replication and facilitates the transition from metaphase to anaphase during cell division [69,70]. While the role of the cohesin complex proteins in regulating chromatin has been well studied, the role of STAG2 in myeloid leukemogenesis has been more challenging to elucidate.

STAG2, alongside the remaining three core cohesin partners—RAD21, SMC1A, and SMC3—promotes hematopoietic stem cell differentiation [71]. Inactivation of any of the four core cohesin complex proteins creates stem cell self-renewal [72,73]; knockout mouse models showed changes strikingly consistent with early leukemogenesis, with induction of acute leukemia following the acquisition of *FLT3*-ITD [74]. Efforts to target *STAG2* mutations in AML, however, have not focused on indirectly targeting common cooperating mutations but on leukemic clone elimination directly through synthetic lethality.

*STAG2* knockout cell lines had homologous recombination deficiencies [75]. Talazoparib, a poly (ADP-ribose) polymerase (PARP) inhibitor, resulted in a significantly increased response in *SRSF2*^mut^ cells compared with wild-type cells [75]. Therefore, DNA damage deficiencies appear to sensitize *STAG2* knockout cells to PARP inhibition. Talazoparib showed moderate anti-leukemic activity as monotherapy in a molecularly unselected, heavily pretreated AML population [76]; combination approaches with decitabine or gemtuzumab ozogamicin suggested combination approaches were safe and tolerable [77,78].

STAG1, a protein paralogous to STAG2, is an additional target for the induction of synthetic lethality in *STAG2*^mut^ AML. Cancer cells depend on the overlapping function of STAG1 in *STAG2* knockout cell lines to maintain the integrity of chromatin cohesion; double knockout of *STAG1* and *STAG2* induces synthetic lethality [79]. Recently, investigators performed a proof-of-concept demonstration in primary human leukemic cells: disrupting STAG1 eliminated *STAG2*^mut^ AML cells and was well tolerated by wild-type hematopoietic stem cells [80]. These findings pave the way for the evaluation of STAG1 inhibition for the treatment of *STAG2*^mut^ AML.

### 3.5. RUNX1

Runt-related transcription factor 1 (RUNX1) mutations do not directly alter chromatin structure. However, they affect the binding of transcription factors to chromatin and indirectly influence gene expression through altered chromatin interactions [81]. *RUNX1* encodes the α subunit of the core-*binding* factor and is therefore responsible for *binding* transcription complexes to DNA [82]. There are two broad categories of *RUNX1* aberrations in AML: *RUNX1*-related chromosomal rearrangements and *RUNX1* somatic mutations. In the most well-known *RUNX1*-related rearrangement in AML, *RUNX1* fuses to a transcriptional corepressor (*RUNX1T1*), creating the t(8;21) fusion protein, *RUNX1::RUNX1T1*, which we have reviewed elsewhere [51]. While this *RUNX1*-related rearrangement in AML confers a favorable prognosis, somatic mutations do not [2]. Therefore, we will now focus on the role of the somatic *RUNX1*^mut^ in AML.

Truncated *RUNX1* mutations lack DNA binding activity; missense mutations similarly lack DNA binding activity and down-regulate wild-type *RUNX1,* creating a dominant-negative effect [81]. Other dominant-negative RUNX1 mutations occupy the target site on DNA and block functional RUNX1 from occupying the same sites [83]. These alterations have profound downstream effects: mutated RUNX1 disrupts a transcription complex that regulates critical targets of hematopoiesis [84]. Although wild-type RUNX1 is necessary for the genesis of core-binding factor leukemia and rearrangements of mixed lineage leukemia AML, functional RUNX1 is also essential for hematopoietic differentiation [84]. In addition to its role in hematopoietic differentiation, RUNX1 also controls cell cycle regulation, the p53 pathway, and ribosome synthesis [84].

The effects of *RUNX1* mutations in AML suggest sensitivity to ribosome disruptions (Figure 4). Investigators are now targeting *RUNX1*^mut^ AML after discovering that the knockdown of wild-type *RUNX1* induces more potent cell death with ribosome inhibitors [85]. Homoharringtonine is an antileukemic ribosome inhibitor derived from a plant alkaloid that shares structural properties with its synthetic form, omacetaxine mepesuccinate. Treatment with homoharringtonine induces a similar gene expression signature as *RUNX1* knockdown [85]. AML blasts with *RUNX1*^mut^ demonstrated synthetic lethality in a combination approach with omacetaxine mepesuccinate and venetoclax, perhaps partly owing to the omacetaxine-related repression of *MCL1*, an antiapoptotic protein commonly implicated in venetoclax resistance [85,86]. A clinical trial evaluating the combination of omacetaxine and venetoclax in *RUNX1*^mut^ AML is recruiting at MD Anderson Cancer Center (NCT04874194).

## 4. Conclusions and Future Directions

The addition of these seven recurrent genetic aberrations to the ELN 2022 adverse risk category has refined the prognostication of AML. This group of newly added mutations plays a dominant role in splicing and epigenetic regulation, with unique opportunities for targeted therapies in each molecular cohort. In addition, alternative therapeutic approaches likely exist in larger subgroups by exploiting the nature of the underlying epigenetic or splicing aberrations.

For example, spliceosome mutations lead to alternative splicing of other splicing factors [37]. This results in an aberrant splicing cascade, creating a diverse array of neoantigens. Investigators have recently proposed that the accumulation of neoantigens could make AML with spliceosome mutations more responsive to antibodies against programmed cell death 1 (PD-1) or its ligand (PD-L1) [37]. Limited sample sizes have precluded definitive analyses of outcomes with immunotherapy; however, investigators reported outcomes in a more general cohort of secondary AML.

A phase II study of pembrolizumab and cytarabine in relapsed or refractory AML resulted in a composite complete remission rate of 25.0% in secondary AML [87]; similarly, the combination of nivolumab and azacitidine produced a composite complete remission rate of 22.0% and superior overall survival compared with historical controls treated with azacitidine [88]. In the nivolumab and azacitidine cohort, 44% of patients in the immunotherapy arm had secondary AML. In Project ERIS, we identified one patient (AML-230) with *ZRSR2*^mut^ AML that underwent 19 cycles of consolidation with nivolumab following conventional 7 + 3; the patient achieved a complete response, and the overall survival was 26.3 months. Nevertheless, reports of immunotherapy-associated outcomes in AML with spliceosome mutations are limited.

Other approaches indirectly target splicing. RBM39, an RNA splicing factor, prolongs the survival of AML cells, suggesting a role in AML maintenance [89]. E7820 is an aryl sulfonamide that degrades RBM39 [90]. It induces aberrant splicing and disrupts downstream targets responsible for AML maintenance, such as *HOXA9*, resulting in the preferential lethality of spliceosome-mutated AML [89]. In a phase II trial, investigators administered a similar aryl sulfonamide, indisulam, in combination with idarubicin and cytarabine as salvage therapy in MDS and AML. The cohort was molecularly unselected; the response rate was 35%, and the median duration of response was 5.3 months [91]. Other targeted approaches in spliceosome-mutated AML include disrupting post-translational splicing factors or areas of replication stress, known as R-loops [90,92]. The effectiveness of indirect approaches remains an open question: can we target spliceosome mutations as a general class?

Mutational cooperation invariably complicates targeted approaches. Combination therapies with multiple targeted agents hold promise in the treatment of AML, and clinical trials should be designed to encourage the recruitment of specific molecular cohorts. Innovations in AML require mutation-driven treatment paradigms to bring new hope to the field as we begin a new chapter of the molecular era.

## Figures and Tables

**Figure 1 cancers-15-03292-f001:**
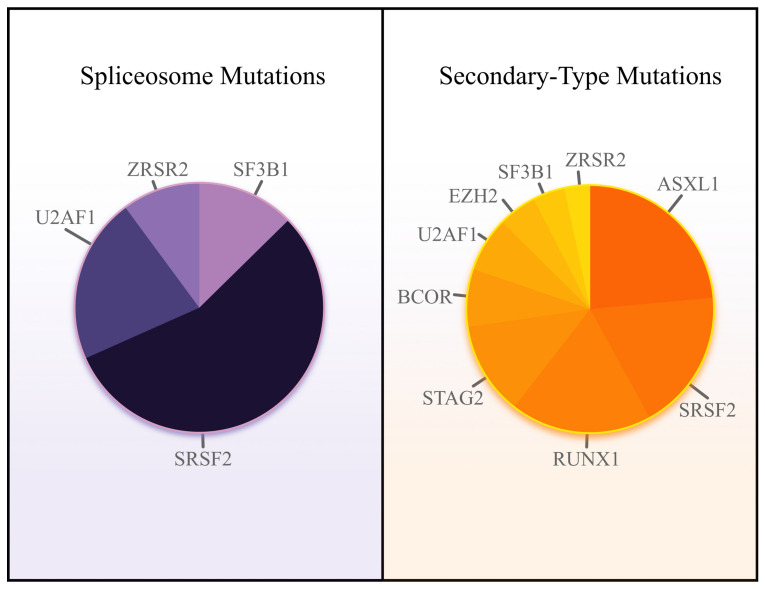
Relative spliceosome mutation frequency (**left**) and secondary-type mutation frequency (**right**) in AML. Data source: VCU Massey Cancer Center Project ERIS database.

**Figure 2 cancers-15-03292-f002:**
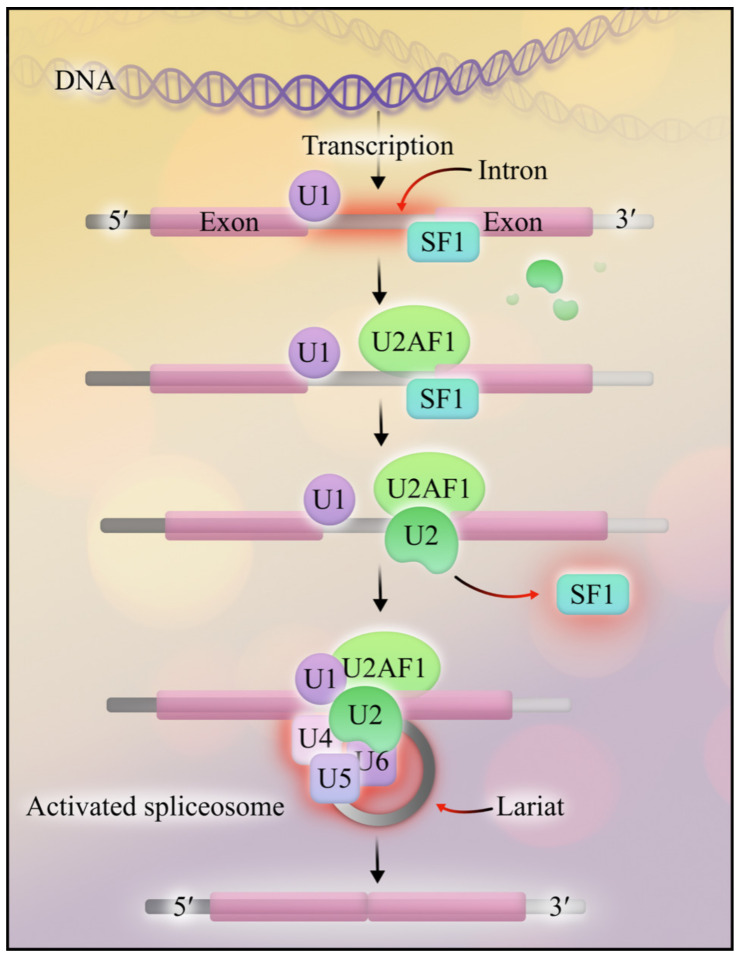
The spliceosome. After mRNA is transcribed from DNA, U1 binds to the 5′ intron region, and SF1 binds to the 3′ region. The U2AF1 complex is recruited near the 3′ region, which facilitates the binding of U2 and displaces SF1. Next, U4, U5, and U6 aggregate to form the activated spliceosome. The spliceosome creates a lariat from the intron loop, which is excised; the exons are ligated through two sequential transesterification reactions.

**Figure 3 cancers-15-03292-f003:**
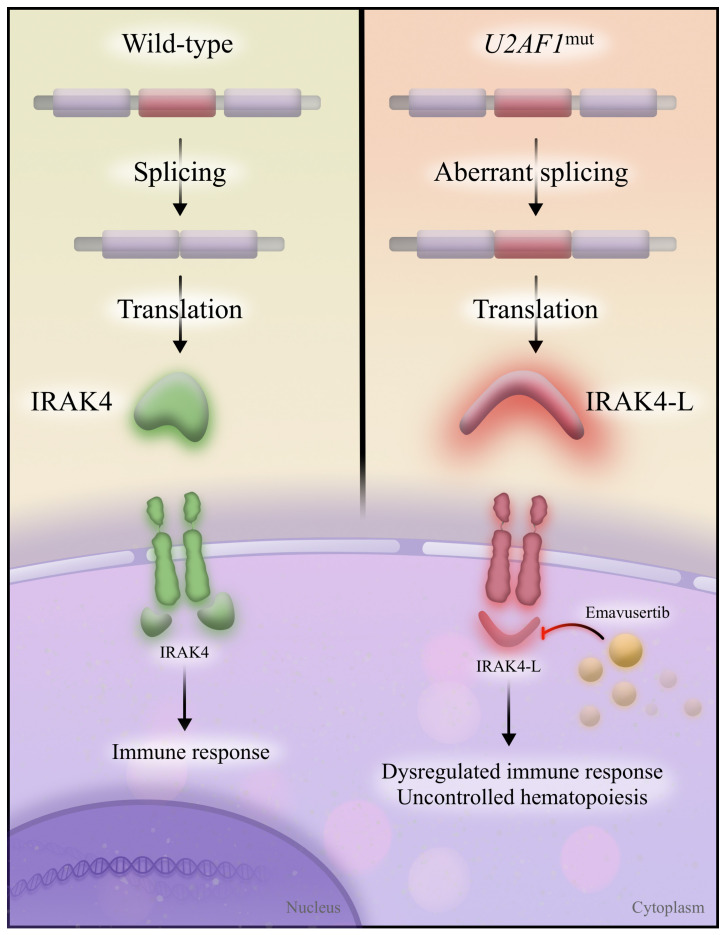
IRAK4 and IRAK4-L. Wild-type spliceosome proteins allow for the correct splicing of IRAK4 (**left**), which removes an exon (red) and creates a functional IRAK4 protein. IRAK4 then appropriately activates the immune response. In the presence of U2AF1 mutations (**right**), the exon is inappropriately included in the final transcript, resulting in a longer IRAK4 isoform (IRAK4-L). IRAK4-L results in dysregulated immune response and uncontrolled hematopoiesis; it is inhibited by emavusertib.

**Figure 4 cancers-15-03292-f004:**
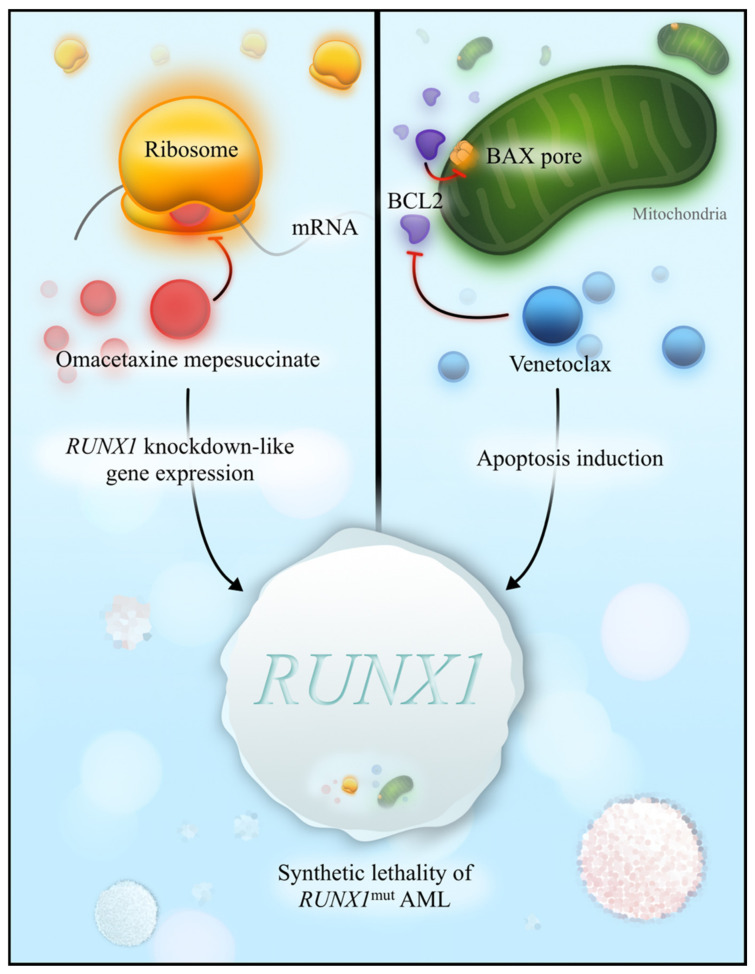
Synthetic lethality in *RUNX1*^mut^ AML. Omacetaxine mepesuccinate is a ribosome inhibitor—it induces *RUNX1* knockdown-like gene expression (**left**). Venetoclax is a BCL2 inhibitor that binds to BCL2 and allows the BAX pore to form on the surface of mitochondria. BAX pore formation facilitates apoptosis. Omacetaxine and venetoclax result in synthetic lethality in *RUNX1*^mut^ AML.

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
