# Peer review of "Secondary-Type Mutations in Acute Myeloid Leukemia: Updates from ELN 2022"

_cancers, 2023, doi:10.3390/cancers15133292_

Round 1

Reviewer 1 Report

In the article “Secondary-type mutations in acute myeloid leukemia” the authors presented novel molecular markers that became part of the new ELN 2022 classification of AML and how they can be used for the design and application of targeted therapy. The text is well written, with adequate up-to-date references. Figures are suitable and well described.

I find that the article deals with a very interesting and useful topic, and I recommend it for publication in Cancers in its current form.

Author Response

Thank you very much.

Reviewer 2 Report

This is an interesting review article by Bouligny and colleagues on secondary-type mutations in AML. The topic is fairly well reviewed but the major focus seems to be on splicing factor mutations and IRAK and FLT3 inhibition with emavusertib. 

The title should be updated to reflect the focus on spliceosome mutations since more than half of the text is dedicated to this. 

Other therapeutics that target the spliceosome are not discussed at all. A few of these are in clinical trials and have data published already such as H3B8800 and E7820. Although these studies included both MDS and AML there were AML patients treated. 

Can the authors please provide a reference for the ZRSR2 part of this sentence? "Additionally, leukemia cells with ZRSR2 knockdown exhibited slower growth compared with wild-type controls, a pattern not exclusive to loss of ZRSR2: U2AF1mut AML demonstrated similar findings of reduced hematopoietic reconstitution. Therefore, mutations in ZRSR2 appear to lead to constitutive stem cell self-renewal but do not appear to contribute to proliferative phenotypes" - The study referenced in the prior sentence actually shows increased proliferative advantage of Zrsr2 knockout in competetive transplants in mice. 

Agree with not including TP53 mutations since they appear to be distinct entity but this should be addressed as some readers may be confused as to why TP53 not included in secondary type mutations. 

Author Response

Reviewer 2:

1. This is an interesting review article by Bouligny and colleagues on secondary-type mutations in AML. The topic is fairly well reviewed but the major focus seems to be on splicing factor mutations and IRAK and FLT3 inhibition with emavusertib. 

1. We thank the reviewer for this feedback. Upon review, we recognize we mentioned IRAK4 twice — once in the spliceosome section and a second time in the U2AF1 section. We agree this may have drawn unnecessary focus on emavusertib. Therefore, we removed the section on IRAK4 in the spliceosome overview to simplify the spliceosome section and limit IRAK4 only to the U2AF1 section (section 1.1, paragraph 4).

2. The title should be updated to reflect the focus on spliceosome mutations since more than half of the text is dedicated to this. 

2. We thank the reviewer for this point — we agree that spliceosome mutations are significant entities in secondary AML. As four out of seven of the mutations recently introduced in the ELN 2022 adverse risk category are spliceosome mutations, we felt that a significant focus on the spliceosome would be reasonable. As mentioned above, we did simplify the spliceosome overview text, as requested. While we did consider a title that focuses on spliceosome mutations in line with the reviewer’s suggestion, we ultimately felt that the current title reflects the overall work by not reducing the emphasis on the non-spliceosome mutations.  We sincerely appreciate this suggestion.

3. Other therapeutics that target the spliceosome are not discussed at all. A few of these are in clinical trials and have data published already such as H3B8800 and E7820. Although these studies included both MDS and AML there were AML patients treated. 

3. We appreciate the reviewer for pointing out the spliceosome-targeting drugs. We agree with the reviewer that this drug class is worthy of review — we discussed H3B-8800 in sections 2.1, paragraph 3, sentences 5–6, and in 2.2, paragraph 3, sentences 4–5. Per the reviewer’s suggestion, we added a section on E7820 (section 4, paragraph 4, sentences 1–7). We included a discussion of E7820 in a molecularly unselected phase II trial in combination with intensive chemotherapy. In addition, we also mentioned post-translational splicing factor and R-loop disruption as potential targets in splicing-factor mutated AML. We thank the reviewer for suggesting these additions.

4. Can the authors please provide a reference for the ZRSR2 part of this sentence? "Additionally, leukemia cells with ZRSR2 knockdown exhibited slower growth compared with wild-type controls, a pattern not exclusive to loss of ZRSR2: U2AF1mut AML demonstrated similar findings of reduced hematopoietic reconstitution. Therefore, mutations in ZRSR2 appear to lead to constitutive stem cell self-renewal but do not appear to contribute to proliferative phenotypes" - The study referenced in the prior sentence actually shows increased proliferative advantage of Zrsr2 knockout in competetive transplants in mice. 

4. Thank you for pointing this out; we have added the reference to the end of the sentence that demonstrated that ZRSR2 knockdown leukemia cells showed a general tendency for slower growth than control cells.

5. Agree with not including TP53 mutations since they appear to be distinct entity but this should be addressed as some readers may be confused as to why TP53 not included in secondary type mutations. 

5. Thank you; we also agree that TP53 mutations would likely be out of scope for this review. We have clarified that TP53mut AML represents a distinct category of adverse-risk AML in section 4, paragraph 1, sentence 1.

Round 2

Reviewer 2 Report

The authors have addressed my comments except for the title being too broad. They are correct that 5 out of the 7 new genes introduced in the new ELN adverse risk are spliceosome but then perhaps the title should clarify that the focus is on new updates from ELN 2022. This can be done through a subtitle. As a reader seeing the title secondary-type mutations in AML I would expect to read about TP53 and many others that are not mentioned. We agree that TP53 mutated AML are a different category but this should be clarified in the title since the review is mostly about spliceosome, EZH2 and RUNX1.  

Author Response

Thank you very much; we have revised the title accordingly: "Secondary-Type Mutations in Acute Myeloid Leukemia: Updates from ELN 2022." We appreciate your recommendation.